# Characterization and Vaccine Potential of Outer Membrane Vesicles from *Photobacterium damselae* subsp. *piscicida*

**DOI:** 10.3390/ijms24065138

**Published:** 2023-03-07

**Authors:** Alexandra Teixeira, Inês Loureiro, Johnny Lisboa, Pedro N. Oliveira, Jorge E. Azevedo, Nuno M. S. dos Santos, Ana do Vale

**Affiliations:** 1Fish Immunology and Vaccinology, i3S—Instituto de Investigação e Inovação em Saúde, Universidade do Porto, 4200-135 Porto, Portugal; 2EPIUnit, ICBAS-Instituto de Ciências Biomédicas Abel Salazar, Universidade do Porto, 4050-313 Porto, Portugal; 3Organelle Biogenesis and Function, i3S—Instituto de Investigação e Inovação em Saúde, Universidade do Porto, 4200-135 Porto, Portugal; 4ICBAS-Instituto de Ciências Biomédicas Abel Salazar, Universidade do Porto, 4050-313 Porto, Portugal

**Keywords:** *Photobacteriosis*, extracellular products, OMVs, virulence factors, vaccination

## Abstract

*Photobacterium damselae* subsp. *piscicida* (*Phdp*) is a Gram-negative fish pathogen with worldwide distribution and broad host specificity that causes heavy economic losses in aquaculture. Although *Phdp* was first identified more than 50 years ago, its pathogenicity mechanisms are not completely understood. In this work, we report that *Phdp* secretes large amounts of outer membrane vesicles (OMVs) when cultured in vitro and during in vivo infection. These OMVs were morphologically characterized and the most abundant vesicle-associated proteins were identified. We also demonstrate that *Phdp* OMVs protect *Phdp* cells from the bactericidal activity of fish antimicrobial peptides, suggesting that secretion of OMVs is part of the strategy used by *Phdp* to evade host defense mechanisms. Importantly, the vaccination of sea bass (*Dicentrarchus labrax*) with adjuvant-free crude OMVs induced the production of anti-*Phdp* antibodies and resulted in partial protection against *Phdp* infection. These findings reveal new aspects of *Phdp* biology and may provide a basis for developing new vaccines against this pathogen.

## 1. Introduction

*Photobacterium damselae* subsp. *piscicida* (*Phdp*) is a Gram-negative pathogen that causes a severe septicemic disease in many warm water marine fish species in Europe, Asia, and North America [1]. *Phdp* was first isolated in a massive epizootic that occurred in wild populations of white perch (*Morone americanus*) and striped bass (*Morone saxatilis*) in 1963 in Chesapeake Bay (USA) [2]. Since then, *Phdp* has been isolated in different geographical areas, and has been continuously imposing serious constraints on the aquaculture production of several economically important fish species, including sea bream (*Sparus aurata*) [3], European sea bass (*Dicentrarchus labrax*) [4], and sole (*Solea senegalensis* and *Solea solea*) [5], which are mainly cultured in Mediterranean countries, as well as yellowtail (*Seriola quinqueradiata*) [6] and cobia (*Rachycentron canadum*) [7], primarily produced in Japan and China, respectively. According to the FAO, the production of these species reached 700 thousand tones in 2019 [8].

*Phdp* outbreaks are often associated with high water temperatures (>23 °C) and poor water quality [9] and lead to mortalities that can be as high as 80% of the affected stock [10]. Acute *Phdp* infections are characterized by the occurrence of generalized bacteremia and extensive cytopathology with abundant tissue necrosis [11]. Whitish tubercle-like lesions of 0.5 to 3.5 mm in diameter, consisting of accumulations of bacteria, apoptotic cells, and necrotic debris are frequently present in the spleen and head-kidney of infected fish [9,10,11,12,13]. Despite the high negative impacts caused by this pathogen, only a few *Phdp* virulence factors are presently known and, thus, knowledge on *Phdp* pathogenicity remains incomplete.

Antibiotics were the first approach used to treat and control *Phdp* infections, but *Phdp* became resistant to many antimicrobials [1]. Furthermore, there are huge concerns in using antibiotics to treat infections in aquaculture, because it can result in antibiotic residues in the final product and in the emergence and spreading of antibiotic resistance among bacterial species [14,15,16]. In this context, vaccination emerged as the most promising approach to control *Phdp* disease outbreaks. Indeed, over the last 30 years, several anti-*Phdp* vaccine strategies were proposed and there are presently a few inactivated bacterins commercially available to prevent *Phdp* infections in some species [1,17]. However, these vaccines have variable efficacy [1] and serious outbreaks of *Phdp* continue to occur in several countries [18,19,20], even in vaccinated stocks, underscoring the need for novel and improved vaccines able to prevent this disease.

Studies performed in the 1990s suggested that factors secreted by *Phdp* display important pathobiological activities [13,21]. The specific components responsible for the observed activities, however, remained unidentified for many years. It is now known that *Phdp* secretes large amounts of AIP56, an apoptogenic toxin that targets and destroys host phagocytes and plays a crucial virulence role [11,22,23]. Additionally, it was also reported that *Phdp* secretes PnpA, a peptidoglycan hydrolase that can degrade the peptidoglycan of potential *Phdp* competitors [24], although the role of this protein in *Phdp* virulence remains unknown. In addition to these two factors, which are secreted by the type II secretion system (T2SS) of *Phdp* [24,25], it was recently reported that most isolates of *Phdp* harbor a type III secretion system (T3SS) [26,27], which is encoded in an unstable virulence-associated plasmid [28]. Although it is likely that the T3SS is involved in *Phdp* virulence, its specific functions remain undisclosed [28].

Outer membrane vesicles (OMVs) are nanosized spherical bilayered particles released by Gram-negative bacteria during in vitro culture, as well as in the environment and during host infection [29,30]. It is now recognized that they are involved in several important functions, such as cell-to-cell communication, stress responses, antimicrobial resistance, horizontal gene transfer, and immune evasion [29,30,31]. OMVs originate by the blebbing of the outer membrane and contain biologically active components, such as lipopolysaccharide (LPS), phospholipids, and major outer membrane proteins (OMPs), as well as periplasmic components that are entrapped during vesiculation [29,31,32]. OMVs produced by bacterial pathogens also carry virulence factors and may work as long-distance delivery vehicles that can contribute to host colonization and infection-associated pathology [29,31].

In addition to their physiological and pathogenic roles, OMVs have been receiving increasing attention in the field of vaccinology, because they present bacterial antigens in native conformations and are able to induce potent protective humoral and cell-mediated immunity [29,33,34]. Additionally, they have inanimate activity, non-replicative properties and high stability [29,34], further supporting their use as vaccine components. To date, the most successful examples of the use of OMVs in vaccines are the human anti-*Neisseria meningitidis* group B (MenB) vaccines, which are used to control specific outbreaks of MenB-associated meningitis as well as to protect against the endemic disease in several countries [35,36]. Several other OMV-based vaccines are presently being developed against animal and human pathogens [37,38].

The protection induced by OMVs-based vaccines is thought to be mediated by the large numbers of pathogen associated molecular patterns (PAMPs) present in these vesicles, as PAMPs are known to play a key role in stimulating host innate immunity [29,37]. One of the major advantages of using OMVs as vaccines is that they possess intrinsic adjuvant capacity [29,33], while lacking the side-effects associated with immune-enhancing adjuvants. Another advantage is that OMVs can be engineered to remove irrelevant/deleterious antigens and toxic components and/or to incorporate autologous or heterologous protective antigens, further expanding their applicability for vaccination [39,40]. Despite the significant potential of OMVs for vaccine development and the urgent need for new and improved fish vaccines, there are only a few studies on OMV-based vaccines for fish. A promising result was recently described by Park and colleagues, who showed that the administration of OMVs from the fish pathogen *Edwardsiella tarda* to olive flounder (*Paralichthys olivaceus*) conferred partial protection against experimental edwardsiellosis [41].

In this work, we report that *Phdp* secretes large numbers of OMVs when cultured in vitro and during infection in vivo. These OMVs were purified and the most abundant vesicle-associated proteins were identified. We also show that purified OMVs are able to protect *Phdp* from the bactericidal activity of fish antimicrobial peptides, suggesting that the secretion of OMVs in infected hosts may contribute to the evasion of *Phdp* from the host innate immunity. Finally, vaccination experiments showed that OMVs administered without adjuvant were able to induce a partial, but significant, protection against *Phdp* infection. Although the relative percent survival (RPS) of OMVs-vaccinated fish was relatively low (35–38%), these results are promising and provide the basis for future work aiming at improving the OMVs protective efficacy, for example, by altering their antigenic content.

## 2. Results

### 2.1. Photobacterium damselae subsp. piscicida MT1415 Strain Secretes Large Numbers of OMVs In Vitro

Transmission electron microscopy (TEM) analyses of *Phdp* MT1415 cell suspensions negatively stained with uranyl acetate revealed very large numbers of small spherical vesicles surrounding the bacterial cells (Figure 1A). The size and morphological characteristics of the vesicles suggested that they could represent outer membrane vesicles (OMVs). To test this hypothesis, we fixed *Phdp* MT1415 whole colonies and processed them for conventional TEM. Indeed, the ultrathin sections of these colonies revealed an abundance of vesicles located near the bacterial cells’ surface (Figure 1B).

To further characterize these vesicles, cell-free culture supernatants from late-exponential cultures of the *Phdp* strain MT1415 were concentrated by tangential filtration, dialyzed to eliminate components of the culture medium and soluble secreted proteins, and ultracentrifuged to obtain a crude preparation of vesicles. TEM analysis of this preparation after negative staining revealed a rather homogeneous population of spherical vesicles, most of which with less than 50 nm in diameter (Figure 1C). The observation of the ultrathin sections of the vesicles revealed that they are surrounded by a single bilayer membrane (Figure 1D), supporting the idea that, indeed, they originated from the blebbing of the outer membrane [32]. In agreement with the TEM observations, analyses of OMVs by dynamic light scattering (DLS) showed a quite homogeneous size-distribution, with the majority of the vesicles (>97%) between 16 and 44 nm in diameter (Figure 1E). We used a previously described TEM-based approach [42] to determine the number of vesicles in our preparations. Yields of approximately 1.6x10^12^ OMVs per mL of bacterial culture were routinely obtained.

### 2.2. Identification of the Most Abundant Proteins in Phdp MT1415 OMVs

To identify the main protein components of *Phdp* OMVs, crude preparations of vesicles were subjected to SDS-PAGE and the main protein bands were excised, trypsinized, and subjected to mass spectrometry. The protein profile of crude MT1415 OMVs is shown in Figure 2A, and the identities of the most abundant proteins are listed in Table 1.

As expected, the majority of the identified proteins correspond to proteins predicted to be associated with the outer membrane of *Phdp*, including eight outer membrane proteins (Omp) with molecular weights ranging from 18.8 to 42.3 kDa (bands H-M), a 64.4 kDa TonB-dependent receptor (band E), and two outer membrane lipoproteins (bands N and O). Additionally, the vesicles also contained four uncharacterized proteins, including a putative lipase (band B), an Ig-like domain-containing protein (band C), a domain of unknown function 3466-containing protein (band D), and a protein with homology to the insecticidal cry toxins (band G). In band G, the recently characterized peptidoglycan hydrolase PnpA was also identified [24]. Finally, the vesicles also contained the well-characterized apoptogenic toxin AIP56 (bands A and F) [22,23]. Considering the molecular weight of AIP56 (56.2 kDa), it is likely that band A corresponds to an SDS-resistant AIP56 dimer. BLAST analysis using the identified proteins as queries revealed that, with the exception of AIP56, the most abundant proteins present in *Phdp* OMVs are conserved in the closely related subspecies *Photobacterium damselae* subsp. *damselae* (Table 1).

Previous studies have shown that AIP56 and PnpA are soluble proteins abundantly secreted into the extracellular medium by the T2SS of *Phdp* [24,25]. Neither AIP56 nor PnpA possess transmembrane domains and, therefore, their presence in crude OMVs could result from the entrapment of the periplasmic pools of the proteins during vesicle formation. Alternatively, the small amounts of AIP56 and PnpA found in crude OMVs might simply represent adsorbed material or aggregates that co-sediment with the vesicles. To clarify this, OMVs were further purified by Histodenz density gradient centrifugation. In this method, soluble proteins stay at the bottom of the gradient, whereas proteins strongly associated with lipids float and are recovered in the upper fractions. In the gel shown in Figure 2B, the OMVs-containing fractions (fractions 5–9) can be identified by the presence of several bona fide outer membrane proteins (e.g., bands E, L, and M). In contrast, the bulk of AIP56 and PnpA was recovered in the bottom of the gradient (fractions 10–12 and pellet), suggesting that the majority of AIP56 and PnpA found in crude OMVs are not associated with the vesicles. There was, however, a small pool of both proteins that floated with OMVs. To determine whether these pools are located in the vesicles lumen or represent proteins adsorbed to the outer surface of the vesicles, fractions 5–7 were pooled and subjected to a protease protection assay (Figure 2C,D). OMVs were left untreated or were treated with Triton X-100 (TX-100), incubated with proteinase K (PK), as indicated, and analyzed by SDS-PAGE and Western blotting, using anti-AIP56 and anti-PnpA antibodies. As shown in Figure 2C, in the absence of detergent, the integral membrane proteins TonB-dependent receptor (band E) and Omp proteins (bands H, I, and L) were protected from PK digestion (Figure 2C, lane 2) but were readily degraded by the protease after the solubilization of the OMVs with TX-100 (Figure 2C, lane 3). In contrast, the treatment of intact OMVs with PK resulted in the total degradation of AIP56 (lane 2 in Figure 2C,D—upper panel), indicating that this protein is adsorbed to the external surface of the vesicles. A similar result was obtained for PnpA. Indeed, PnpA is also accessible to PK in intact OMVs (Figure 2D—lower panel, lane 2), although in this case, proteolysis is not complete—the protein is clipped by 1–2 kDa by PK, a property that reflects intrinsic resistance of PnpA to the protease, as assessed by its behavior in detergent-solubilized OMVs treated with the protease (Figure 2D, lower panel, lane 3).

### 2.3. Large Amounts of OMVs Are Also Secreted In Vitro by the Phdp Strain PP3 and by Two Field Isolates of Phdp

To investigate whether the shedding of large numbers of OMVs is a general feature of *Phdp*, we investigated vesicle production in a Japanese strain (PP3) isolated from *Seriola quinqueradiata* and in two field isolates recovered in 2019 from diseased *Dicentrarchus labrax* and *Sparus aurata* in Greece and Spain, respectively (see Table 2). Crude preparations of OMVs from these strains were obtained and analyzed by TEM, DLS, and SDS-PAGE (Figure 3). TEM analyses (Figure 3A) showed that PP3, SPSA19-2, and GRDL19-1 produce OMVs morphologically similar to the ones obtained from the MT1415 strain (see Figure 1) and DLS revealed a similar size distribution for the OMVs of all strains/isolates (Figure 3B). Additionally, SDS-PAGE revealed that the most abundant OMVs-associated proteins are common to all four strains (Figure 3C). There are, however, slight differences between the polypeptide compositions of OMVs from the different strains, particularly in SPSA19-2 and PP3. The identities of the extra minor protein bands present in the OMVs of these strains remain unknown. Nevertheless, these results indicate that the production of large numbers of OMVs in vitro is not a strain-specific characteristic but rather a general property of *Phdp*.

### 2.4. Phdp Releases OMVs during Infection In Vivo

The observation that *Phdp* releases large amounts of OMVs during in vitro culture prompted us to investigate whether the same is true in vivo during host infection. To achieve this, ultrathin sections of spleens collected from sea bass infected with the *Phdp* field isolate GRDL19-1 were analyzed by conventional TEM. The spleen was selected for this analysis because it is one of the most affected organs in *Phdp* infections [11,43]. As shown in Figure 4A–D, large numbers of bacteria were observed in the spleen sections of infected fish. Analyses of the samples at higher magnifications revealed the presence of many small vesicles in the vicinity of bacteria, some of which seem to be budding out from the bacterial cells (Figure 4(A1–D1,A2–D2)). These results suggest that *Phdp* releases OMVs in vivo in infected hosts.

### 2.5. OMVs Protect Phdp from the Bactericidal Activity of Fish Antimicrobial Peptides

Previous work with Gram-negative pathogens have shown that OMVs can protect bacterial cells from noxious environmental agents that target the bacterial outer membrane, such as antibiotics, bacteriophages, and antimicrobial peptides (AMPs) [44,45,46,47,48]. Considering that fish AMPs have key roles in the innate immune response and are crucial for the first line of defense against a wide variety of pathogens [49,50], we asked whether *Phdp* OMVs could protect the bacteria from the bactericidal activity of piscidin 1 and piscidin 2, two recently characterized antimicrobial peptides from sea bass [51]. For this, we incubated *Phdp* MT1415 cells in TSB-1 containing the inhibitory concentrations of sea bass piscidin 1 or piscidin 2, in the presence or absence of OMVs, and evaluated bacterial growth by measuring the optical density at 600 nm (OD_600nm_) overtime and by plating the serial dilutions of the cultures in TSA-1. As expected, in the absence of AMPs, typical growth curves were obtained in the TSB-1 medium, whereas a complete growth inhibition was observed in TSB-1 supplemented with 7.4 or 9.1 µM of piscidin 1 and piscidin 2, respectively (Figure 5). Importantly, when the medium containing piscidin 1 or piscidin 2 was supplemented with crude OMVs prior to inoculation, the bacteria were able to grow (Figure 5), indicating that *Phdp* OMVs are able to protect *Phdp* cells from the bactericidal activity of host AMPs.

### 2.6. Vaccination of Sea Bass with OMVs Confers Partial Protection against a Phdp Challenge

To evaluate the vaccine efficacy of *Phdp* OMVs, we performed vaccination trials in European sea bass. Since the crude preparations of OMVs from wild type *Phdp* cannot be used for vaccination due to the presence of significant amounts of AIP56, an apoptogenic protein highly toxic for fish [22,23,25,52], we generated MT1415^pPHDP10-^, an MT1415-derived strain cured of the *aip56*-encoding plasmid pPHDP10 [23]. Analysis by TEM (Figure 6A) and DLS (Figure 6B) confirmed that the OMVs produced by wild type MT1415 and MT1415^pPHDP10-^ strains were morphologically similar, and SDS-PAGE and western blotting analysis revealed a similar protein profile, except for the lack of AIP56 in MT1415^pPHDP10-^ OMVs, as expected (Figure 6C).

To assess the vaccine potential of *Phdp* OMVs, two independent trials were performed, using two different lots of sea bass. Fish received two injections of either MT1415^pPHDP10-^ OMVs in PBS, or PBS alone, as control, and were subsequently challenged with the MT1415 strain. As shown in Figure 7A, in both trials, the survival rate in the OMVs-vaccinated group was significantly higher than that in the PBS control, resulting in RPS values of 35.1 and 38.2% for trials I and II, respectively. In agreement with this, sera from fish immunized with OMVs have increased levels of anti-MT1415 antibodies (Figure 7B). Interestingly, the antibodies induced by vaccination with MT1415 OMVs were also able to bind cells from *Phdp* GRDL19-1, a recent field isolate from Greece (see Table 2). Altogether, these results indicate that immunization with *Phdp* OMVs leads to production of antibodies able to bind *Phdp* cells and confers partial protection against *Phdp* infection.

## 3. Discussion

In this work, we show that *Phdp* secretes large numbers of OMVs in vitro and in vivo and report the isolation and detailed characterization of these vesicles. We further show that *Phdp* OMVs are able to protect *Phdp* cells from the microbicidal activity of host AMPs. Data suggesting that *Phdp* OMVs are promising antigens for developing anti-*Phdp* vaccines are also provided.

Crude preparations of OMVs were obtained from *Phdp* culture supernatants using a simple protocol amenable to scaling up. The TEM and DLS analyses of these preparations showed that they contain a homogeneous population of vesicles with sizes ranging from 16–44 nm in diameter. The proteomic analysis of crude OMVs from strain MT1415 revealed that the most abundant proteins present in OMVs are proteins predicted to be components of the outer membrane, including eight members of the Omp family, a TonB-dependent receptor, and two small molecular weight lipoproteins. Our data suggest that the major proteins identified in MT1415 OMVs are also the most abundant ones in the OMVs obtained from PP3 and two field isolates of the bacterium. However, a few additional minor protein bands were exclusively detected in the OMVs from SPSA19-2 and PP3, suggesting that the OMVs’ protein repertoire may vary amongst strains. A detailed proteomic characterization of the OMVs from different field isolates is needed, not only to define the core proteome of *Phdp* OMVs (common to all strains) but also to identify proteins specifically present in OMVs from field strains that may correspond to hitherto unidentified virulence factors. Interestingly, with the exception of AIP56, all proteins identified in *Phdp* OMVs are conserved in the closely related subspecies *Photobacterium damselae* subsp. *damselae* (*Phdd)*, a pathogen of marine animals, including fish, mollusks, crustacea, and cetaceans that can also cause opportunistic infections with fatal outcomes in humans [53]. Whether *Phdd* also secretes large numbers of OMVs remains to be investigated.

The biological functions of many of the proteins contained in *Phdp* OMVs are currently unknown. Evidently, additional work is required to determine whether they have some role in *Phdp* pathogenicity, and if so, to disclose their pathogenic mechanisms. In particular, it would be interesting to investigate the biological roles of the OmpA-like proteins identified in *Phdp* OMVs, since studies performed in several Gram-negative bacteria have shown that OmpA-like proteins are often important for the pathogenesis of infection, including interaction with host tissues to promote host adhesion and invasion, evasion from host defense mechanisms, and the modulation of immune cells’ functions [54,55]. Likewise, it would be interesting to investigate the biological role of the Ig-like domain containing protein identified in *Phdp* OMVs, because it has been shown that several surface-associated bacterial Ig-like domain-containing proteins work as adhesins with key roles in host tissue colonization and infection [56].

Our data show that the secretion of OMVs by *Phdp* occurs not only during in vitro growth conditions but also during infection in vivo, suggesting that they may actively participate in the establishment of infection. While the OMVs produced in vivo look morphologically very similar to those produced in vitro, their detailed characterization is a relevant aspect worth pursuing in future studies. Amongst the many important biological roles attributed to OMVs, it has been proposed that they may facilitate bacterial evasion from the host immune response by functioning as decoys to protect bacterial cells from the microbicidal activity of host AMPs [46]. Piscidins are fish-specific AMPs produced by several species, including European sea bass [51], a species highly susceptible to *Phdp*. Piscidins have a crucial role in innate immunity by acting as a first line of defense against pathogens [50,57]. Amongst sea bass piscidins, piscidins 1–5 have been shown to possess antimicrobial activity against many Gram-negative and Gram-positive bacterial pathogens, including *Phdp* [51], suggesting that they may constitute an important tool used by sea bass to control bacterial infections. In this work, we found that OMVs isolated from the culture supernatants of *Phdp* are able to protect *Phdp* cells from the bactericidal action of sea bass piscidin 1 and piscidin 2 in vitro. Based on this observation, we propose that the OMVs produced by *Phdp* during host infection may allow the pathogen to counteract the antimicrobial activity of the piscidins, thereby facilitating the spreading of the bacteria and the establishment of infection.

Vaccination constitutes the best way to control fish diseases, allowing not only the reduction of economic losses associated with infectious diseases but also the limiting of the extensive use of antibiotics in fish farming that leads to the spreading of antibiotic resistance among bacterial species, some of which are pathogenic for humans [58]. Given the promising characteristics of OMVs for vaccination, we decided to investigate the vaccine potential of *Phdp* OMVs. Since crude OMVs from wild type *Phdp* could not be used for immunization due to the presence of the AIP56 toxin, we generated an MT1415-derived strain that lacked the *aip56*-encoding plasmid pPHDP10 [23] which was used to obtain OMVs devoid of AIP56. The immunization of sea bass with these AIP56-negative OMVs induced the production of anti-*Phdp* antibodies able to recognize cells not only from the MT1415 strain but also from GRDL19-1. In agreement with the increased anti-*Phdp* antibodies levels, immunization with OMVs resulted in a significant protection against experimental infection with *Phdp* MT1415, with RPS values ranging from 35–38%. This protection, albeit moderate, is promising, particularly because OMVs were administered without adjuvant. This is a major advantage for fish vaccination, because it allows the overcoming of the deleterious side effects of oil-based adjuvants that affect fish welfare and compromise the quality of fish products [59,60,61]. Nevertheless, additional studies are required to further explore the vaccine potential of *Phdp* OMVs. In this context, methodologies to manipulate the antigen content of the OMVs, already developed for other bacteria [39,40], are worth exploring in order to remove irrelevant/deleterious antigens and to incorporate components relevant for protection, as this may result in increased protective efficacy.

## 4. Materials and Methods

### 4.1. Bacterial Strains and Culture Conditions

The *Photobacterium damselae* subsp. *piscicida* (*Phdp*) strains used in this study are listed in Table 2. All strains were cultured at 22–25 °C in tryptic soy broth (TSB) or on tryptic soy agar (TSA) (both from Difco^TM^, Becton, Dickinson and Company, Franklin Lakes, NJ, USA) supplemented with NaCl to a final concentration of 1% (*w*/*v*) (TSB-1 and TSA-1, respectively). Stocks of bacteria were maintained at −80 °C in TSB-1 supplemented with 15–30% (*v*/*v*) glycerol.

### 4.2. Generation of a pPHDP10-Negative MT1415 Strain

To obtain AIP56-negative OMVs, we generated an MT1415 derivative lacking the *aip56*-encoding plasmid pPHDP10 [23], because despite several attempts, we were unable to generate an isogenic *Phdp* mutant lacking the *aip56* gene. The pPHDP10 plasmid was eliminated from the MT1415 strain using the curing agent acriflavine. MT1415 was cultured in TSB-1 at 22 °C, overnight (ON) with shaking (160 rpm) and was re-inoculated at 1:200 dilution in TSB-1 with 300 ng/mL acriflavine. Following ON growth at 22 °C with shaking, the culture was re-inoculated in fresh TSB-1 with acriflavine. This last step was repeated until day 16. At each day, an aliquot of the ON culture was removed, serial-diluted, and plated on TSA-1 in order to obtain plates with approximately 200 colonies. The screening of the AIP56+ and AIP56- clones was performed by a blot assay. A nitrocellulose membrane was placed on the plate and incubated at 22 °C for 2 h. The membrane was carefully removed and probed with an anti-AIP56 rabbit antibody [23]. Colonies negative for AIP56 were selected for further testing. The clone used in this study (MT1415^pPHDP10-^*)* was isolated at day 13. The absence of pPHDP10 was confirmed by the agarose gel electrophoresis of plasmid DNA, absence of the AIP56 gene by PCR using the primers pair AIP_FW:5′-GCATGACAGCAATATTTTCT-3′ and AIP_REV: 5′-TTAATTAATGAATTGTGGCG-3′, and the lack of AIP56 protein expression by the SDS-PAGE/Western blotting analysis of bacterial culture supernatants.

### 4.3. Isolation of OMVs

Bacteria from −80 °C stocks were cultured on TSA-1 for 24–72 h at 25 °C, resuspended in TSB-1 at an OD_600nm_ of 0.5–0.6, inoculated 1:100 in 100 mL of TSB-1, and grown for an additional 24 h at 25 °C with shaking (160 rpm). This culture was then used to inoculate 6 × 800 mL of TSB-1 (in 2 L-Erlenmeyer flasks) at 1:100 dilution, followed by incubation at 25 °C with shaking until an OD_600nm_ of 0.9–1.0 was reached. The culture was then centrifuged at 3993× *g* for 30 min at 4 °C (Beckman Avanti J-26 XP, rotor JLA-8.1000), the pellet was discarded, and the supernatant was centrifuged again. The supernatant of this centrifugation was then filtered through a 0.22 µm pore-size filter to remove any remaining bacterial cells. Cell-free supernatant (~4.5 L) was concentrated 45-fold by tangential filtration (Vivaflow 200, 100,000 MWCO PES, VF20H4, Sartorius, Goettingen, Germany); dialyzed against 20 mM Tris pH 8.0, 150 mM NaCl, using the same system; and ultracentrifuged ON at 175,000× *g*, 4 °C (Beckman Optima XE 100, rotor SW 32 TI; Ultra-Clear Tubes, 344058, Beckman Coulter, Brea, CA, USA). The supernatant was discarded and the pellet (crude OMVs) was resuspended in 1.2 mL 0.9% NaCl, aliquoted, frozen in liquid nitrogen, and stored at −80 °C. The sterility of the crude OMV preparations was confirmed by plating aliquots on TSA-1.

### 4.4. Density Gradient Separation

Crude OMVs in floating buffer (20 mM Tris pH 8.0, 150 mM NaCl, 0.5 mM EDTA) containing 50% (*w*/*v*) Histodenz (D2158, Sigma Aldrich) (2.5 mL final volume) were placed at the bottom of a 13 mL centrifuge tube and overlayed sequentially with 5 mL and 2 mL of 40% and 20% Histodenz in floating buffer, respectively, and lastly with 2 mL of floating buffer with no Histodenz. The tube was centrifuged at 160,000× *g*, ON, in a SW 41 TI swing-out rotor (Ultracentrifuge Beckman Optima L80-XP) at 12 °C. After centrifugation, 11 fractions of 1 mL were collected from the top to the bottom of the gradient using a needle connected to a peristaltic pump. The remaining solution (~0.5 mL) was collected as fraction 12. Fractions were frozen in liquid nitrogen and stored at −80 °C.

### 4.5. Transmission Electron Microscopy

For negative staining, 7 µL of crude OMVs diluted 1:100 in 0.9% NaCl or of *Phdp* cells suspension (obtained by culturing *Phdp* on TSA-1 for 48 h at 25 °C followed by resuspension in 1% NaCl) were loaded in a Formvar-carbon coated electron microscopy grid and left to adsorb for 5 min. Excess liquid was gently removed with filter paper and 7 µL of 1% uranyl acetate were applied to the grid and left for 1 min. Excess liquid was removed as above, and the grids were allowed to air-dry.

For the conventional electron microscopy of bacteria and OMVs, *Phdp* colonies obtained by growing on TSA-1 for 24 h or crude OMVs pellets were fixed in 2.5% glutaraldehyde, 2% paraformaldehyde in 0.1 M sodium cacodylate buffer (pH 7.4) for 24 h, post fixed in 1% osmium tetroxide in 0.1 M sodium cacodylate buffer, washed in water, stained with 1% uranyl acetate, dehydrated, and embedded in Embed-812 resin (Electron Microscopy Sciences). Ultrathin sections (40–60 nm thickness) were prepared on a RMC Ultramicrotome (PowerTome, Tucson, AZ, USA), mounted on mesh copper grids, and contrasted with uranyl acetate substitute and lead citrate (both from Electron Microscopy Sciences).

Spleen and head-kidney fragments collected from fish infected with the GRDL19-1 field isolate were processed for conventional electron microscopy following the protocol described above for bacterial cells and OMVs, except that the fixation time was 48 h and the post-fixation was performed with 1% osmium tetroxide and 1.5% potassium ferrocyanide in 0.1 M cacodylate buffer.

All sections were observed under a JEOL JEM 1400 transmission electron microscope at an acceleration voltage of 80 kV. Images were digitally recorded using a CCD digital camera Orius 1100W (Gatan Inc., Pleasanton, CA, USA).

### 4.6. Dynamic Light Scattering

DLS measurements were performed at 25 °C using a Zetasizer Nano ZS (Malvern Instruments, Malvern, UK) equipped with a 4 mW HeNe laser beam with a wavelength of 633 nm and a scattering angle of 173°. Data were collected and analyzed using the Zetasizer Software version 7.13. Size measurements were conducted according to the manufacturer’s instructions, using the “General Mode” analysis model. For data analyses, the viscosity and refractive index (RI) of 0.9% NaCl solution at 25 °C (0.9024 cP and 1.332, respectively) were used.

Crude OMVs were diluted 1:100 (*v*/*v*) in 0.9% NaCl, and 70 µL of the suspension were added to a single-use polystyrene cuvette with a 10 mm path length (ZEN0040). Results were automatically calculated by the software using the Stokes–Einstein equation, and values are reported as the average of three individual measurements. Results were plotted as size distribution by number.

### 4.7. SDS-PAGE and Western Blotting

SDS-PAGE was performed using the Laemmli discontinuous buffer system [62] as described in [23]. OMVs were incubated for 5 min at 95 °C in gel loading buffer (50 mM Tris-HCl pH 8.8, 2% SDS, 0.05% bromophenol blue, 10% glycerol, 2 mM EDTA, and 100 mM DTT) and subjected to electrophoresis in 14 or 15% denaturing polyacrylamide gels. Proteins in the gels were stained with Coomassie Brilliant Blue R-250 or electroblotted onto nitrocellulose membranes. Transfer efficiency and protein loading were assessed by Ponceau S staining.

For Western blotting, membranes were blocked in Tris buffered saline (TBS) containing 5% low-fat powdered milk and 0.1% (*v*/*v*) Tween 20 (TBS-T) for 30–60 min at RT followed by incubation with a rabbit anti-AIP56 [23] or a quail anti-PnpA [24] primary antibody diluted 1:5000 or 1:10,000 in blocking buffer, respectively. Reactive bands were detected using anti-rabbit IgG or anti-chicken IgY secondary antibodies conjugated to alkaline phosphatase (Sigma Aldrich, A9919, and A9171, respectively) diluted 1:10,000, followed by BCIP/NBT development.

### 4.8. Protein Quantification

The total protein content of crude OMVs was determined using the bicinchoninic acid (BCA)-protein assay kit (Pierce, Rockford, IL, USA), following the manufacturer’s instructions, with the addition of 2% SDS [63]. Briefly, prior to protein quantification, the vesicles were solubilized with 2% (*w*/*v*) SDS for 10 min at RT, followed by 1 h at 65 °C. Samples were cooled down at RT for 5–10 min prior to protein quantification. BSA standards were treated similarly.

### 4.9. Proteomic Analysis

Selected protein bands were excised from Coomassie Blue stained SDS-PAGE gels of MT1415 OMVs, reduced with DTT, alkylated with IAA as previously described [64], and digested with trypsin. The resulting peptides were desalted and concentrated using reverse phase C18 tips (ZipTips, Millipore) following the manufacturer’s instructions, eluted in 60% ACN/0.1% TFA, and allowed to dry (SpeedVac, Thermo Scientific, Waltham, MA, USA). The resolubilized peptides were analyzed by MALDI-TOF/TOF mass spectrometry or by liquid chromatography (LC) coupled to an Orbitrap Q-Exactive mass spectrometer (Thermo Scientific) using a nano spray ionization source (Easy-Spray, Thermo Scientific). The peptide mass fingerprint protein identification approach was used for the data acquired by MALDI-TOF/TOF as previously described [65]. The raw data was analyzed with the Mascot software (v 2.6.1 Matrix Science, London, UK) using the UniProt database for the taxonomic selection of *Photobacterium* (May 2017 release). In LC-Orbitrap, reverse phase peptide separation was performed with an Ultimate 3000 system (Thermo Scientific) fitted with a trapping cartridge (Acclaim PepMap C18 100Å, 5 mm × 300 μm i.d., 160454, Thermo Scientific) in a mobile phase of 2% ACN, 0.1% FA at 10 μL min^−1^. After 3 min of loading, the trap column was switched in-line to a PepMap RSLC C18 EASY-Spray analytical column (Thermo Scientific). Separation was generated by mixing A: 0.1% FA and B: 80% ACN, 0.1% FA. Data acquisition was controlled by Xcalibur 4.0 and Tune 2.8 software (Thermo Scientific). The mass spectrometer was operated in data-dependent (dd) positive acquisition mode alternating between a full scan (*m*/*z* 300–2000) and the subsequent HCD MS/MS of the 10 most intense peaks from the full scan (normalized collision energy of 27%). ESI spray voltage was 1.9 kV. Global settings: lock masses best (*m*/*z* 445.12003), lock mass injection Full MS, chrom. peak width (FWHM) 15 s. Full scan settings: 70 k resolution (*m*/*z* 200), AGC target 3e6, maximum injection time 100 ms. dd settings: minimum AGC target 1e3, intensity threshold 1e4, charge exclusion (+) unassigned, 1, 5–8, >8, peptide match preferred, exclude isotopes on, dynamic exclusion 20 s. MS2 settings: microscans 1, resolution 17.5 k (*m*/*z* 200), AGC target 1e5, maximum injection time 100 ms, isolation window 2.0 *m*/*z*, isolation offset 0.0 *m*/*z*, spectrum data type profile. The raw data were processed using Proteome Discoverer software (Thermo Scientific) and searched against the MT1415 entry deposited in NCBI under the accession number PRJNA534205. The Sequest HT search engine was used to identify tryptic peptides. The ion mass tolerance was 10 ppm for precursor ions and 0.02 Da for fragment ions. The maximum allowed missing cleavage sites were set to 2. Cysteine carbamidomethylation was set as a constant modification. Methionine oxidation and N-terminal protein acetylation were defined as variable modifications. For bands in which multiple proteins were identified with statistical confidence (i.e., with at least two unique peptides identified), the relative abundance of each protein was calculated by dividing its abundance value by the sum of the abundance values of all identified proteins.

### 4.10. Protease Protection Assay

For protease protection assay, a pool of fractions 5, 6, and 7 collected after density gradient separation was used. OMVs were left untreated (input control) or treated with 100 µg/mL proteinase K (PK) (Roche Applied Science) for 1 h at RT, with or without prior incubation with 1% (*w*/*v*) Triton X-100 (TX-100) for 15 min at 4 °C. PK was inactivated with phenylmethylsulphonyl fluoride (Roche Applied Science) at a final concentration of 500 µg/mL. TX-100 was added to the samples at 1% final concentration and proteins were precipitated with TCA, as previously described [23], and analyzed by SDS-PAGE/Coomassie Blue staining/Western blotting.

### 4.11. Determination of OMVs Concentration

The concentration of vesicles in crude OMVs preparations was determined using a previously described electron-microscopy approach [42], with some modifications. A 2% (*w*/*v*) methylcellulose (25 cP, Sigma-Aldrich) solution was prepared as described in [66] and diluted in distilled water to obtain a stock solution of 0.4%. To prepare the calibration solution, 20 nm NanoXact™ gold nanospheres (7.4 × 10^11^ particles/mL in 2 mM sodium citrate solution, nanoComposix, San Diego, CA, USA) were centrifuged at 8000× *g* for 2 h at 4 °C, the supernatant was discarded, and the particles were resuspended in an equal volume of 0.4% methylcellulose in order to maintain the particle concentration, i.e., 7.4 × 10^11^ particles/mL. The methylcellulose and calibration solutions were stored at 4 °C. A stock solution of 0.3% uranyl acetate in distilled water was prepared, filtered through a 0.22 µm pore-size filter, and stored at RT.

For quantifying OMV concentrations, a mixture containing 1.11 × 10^11^ gold particles/mL and crude OMVs was freshly prepared by mixing 52.5 µL distilled water, 12.5 µL 0.4% methylcellulose, 10 µL crude OMVs diluted 1:2000 in 0.4% methylcellulose, 15 µL calibration solution, and 10 µL 0.3% uranyl acetate. A drop of 0.5 µL of this mixture was loaded onto square pattern of 100 mesh copper grids and air-dried. OMVs concentration was determined in three different batches of crude MT1415 OMVs, and for each batch, duplicate grids were prepared. Preparations were observed under a JEOL JEM 1400 transmission electron microscope at a magnification of 60,000× and digital images from four non-adjacent squares were acquired, with at least 20 pictures for each square, using a CCD digital camera Orius 1100W (Gatan Inc., Pleasanton, CA, USA). Images from each square were visually inspected using ImageJ [67] and OMVs and gold particles counted manually until reaching at least 200 OMVs. The total number of OMVs and gold particles counted in each square were used to estimate the vesicles concentration in the initial crude OMVs preparation. The concentration of OMVs in each batch was expressed as the mean ± SD of the concentrations obtained for the four squares of the two duplicated grids.

### 4.12. AMPs Protection Assay

Sea bass piscidin 1 (Pisc 1) and piscidin 2 (Pisc 2) [51] were synthesized at NZYtech with 91.45 and 93.08% purity, respectively. MT1415 cultured on TSA-1 for 48 h at 25 °C was suspended in TSB-1 at an OD_600nm_ of 0.5 (corresponding to approximately 6 × 10^8^ CFUs/mL). This suspension was inoculated 1:100 in TSB-1, TSB-1 containing inhibitory concentrations of Pisc 1 (7.4 µM) or Pisc 2 (9.1 µM), or TSB-1 containing the same concentrations of Pisc 1 or Pisc 2 but supplemented with 0.5 or 2 µL crude MT1415 OMVs. Aliquots of the 50 µL of the inoculated media were transferred to wells of a 96-well clear flat bottom plate (Thermofisher), in triplicate, and growth curves were obtained by measuring the OD_600nm_ during 18 h at 20 min interval in a BioTek Synergy 2 spectrofluorometer (BioTeK U.S., Winooski, VT, USA) operating at 25 °C with continuous slow shaking. At the end of the incubation, an aliquot from each well was removed, subjected to serial 10-fold dilutions, and 5 µL of each dilution was plated on TSA-1 and incubated at 25 °C for 24–48 h.

### 4.13. Fish

Two different groups of sea bass (*Dicentrarchus labrax*) with no previous history of *Phdp* infection were used in this study. The first group was used for vaccination trial I and the second for vaccination trial II and for experimental infection with GRDL19-1. The fish were purchased from commercial hatcheries and maintained in 600 L seawater aquaria at a temperature of 20 ± 2 °C, salinity of 23–28‰, and a photoperiod of 14 h light:10 h dark. Water quality was maintained with mechanical and biological filtration and ozone-disinfection. The fish were fed on commercial pellets (Skretting), according to the supplier’s recommendations. During at least 15 days after receiving the fish, their health status was monitored by observing external appearance, swimming behavior, and appetite. The intra-peritoneal (i.p.) injection of vaccines and bacterial inocula was performed under anesthesia achieved by immersion in 0.03% (*v*/*v*) 2-fenoxyethanol, and euthanasia was performed by immersion in 0.06% (*v*/*v*) 2-fenoxyethanol, followed by bleeding. Blood was collected under anesthesia, and euthanasia was applied immediately after blood collection. Organs were collected post-mortem. All experimental protocols involving fish were carried out in accordance with European and Portuguese legislation for the use of animals for scientific purposes (Directive 2010/63/EU; Decreto-Lei 113/2013) and were licensed by the DGAV (Lic. 0421/000/000/2021).

### 4.14. Vaccination Trials

Two independent vaccination trials were performed, using two different lots of fish (average weight of 28.4 ± 5.3 g and 7.3 ± 1.4 g for Trials I and II, respectively). For each trial, fish were distributed into two 200 L tanks (*n* = 60 fish per tank). Vaccination was performed, in a blind fashion, by i.p. injection with 100 µL PBS containing 10 µL of crude MT1415^pPHDP10-^ OMVs (corresponding to approximately 25 µg of protein) or with PBS (as control). After 35–41 days at 20 °C, fish received a booster injection with the same dosage. Twenty fish per group were euthanized at 13–14 days after second immunization to collect blood for measuring anti-*Phdp* antibody levels. Seventeen to nineteen days after second immunization, the remaining 40 fish per group were experimentally infected with the MT1415 strain, as described in Section 4.15, to assess vaccine efficacy. At challenge day, water temperature was raised from 20 ± 2 °C to 25 ± 1 °C. Injected fish were monitored for 10–16 days, at least four times a day, and mortalities were recorded. Any fish showing signs of advanced disease (darkening of body color, lethargy, erratic swimming, and loss of equilibrium) were euthanized and counted as dead. The cumulative probabilities of survival were analyzed using the Kaplan–Meier survival curves and group comparisons were performed using the log-rank test. Differences between groups were considered statistically significant when *p* < 0.05. Vaccine efficacy was evaluated by determining the relative percent survival (RPS), as follows: RPS = [1− (% mortality in vaccinated group/% mortality in control group)] × 100.

### 4.15. Experimental Infections

For assessing vaccine efficacy, fish were i.p.-infected with the MT1415 strain. Bacterial doses used for challenge in Trials I and II were 3.9 × 10^7^ and 4.5 × 10^6^ CFUs/fish, respectively. To collect organs for investigating the production of OMVs in vivo, fish were i.p.-infected with 7.0 × 10^4^ CFUs/fish of the field isolate GRDL19-1.

To prepare the inocula for sea bass infection, bacteria were grown on TSA-1 for 48 h at 25 °C, resuspended in TSB-1 to an OD_600nm_ of 0.5, inoculated 1:100 in TSB-1 and grown for 15–16 h at 25 °C, 160 rpm, until reaching an OD_600nm_ of 0.9–1.0. The culture was centrifuged at 3220× *g*, 20 min, at 4 °C, the pellet resuspended in TSB-1 at an OD_600nm_ of 0.7–0.8 and diluted to achieve the desired bacterial concentration. Injected doses were confirmed by plating serial dilutions of the inocula on TSA-1.

### 4.16. Assessment of Anti-Phdp Antibody Levels

Fourteen or thirteen days after the second immunization in Trials I and II, respectively, 20 fish per group were euthanized, blood collected from the caudal vein, allowed to clot for 24 h at 4 °C, and centrifuged at 5000× *g*, 5 min, 4 °C. Sera were collected and stored at −80 °C.

The content of anti-*Phdp* antibodies in sera were determined by ELISA. Briefly, 96-well micro-titer plates were coated with 50 µL of UV-killed MT1415 or GRDL19-1 cells suspended in 1% NaCl at an OD_600nm_ of 0.8, corresponding to approximately 10^8^ bacterial cells/mL. Wells were blocked with 200 µL of 2% BSA in phosphate-buffered saline (PBS) containing 0.1% Tween 20 (PBS-T) and were incubated for 2 h at RT with 50 µL of the serum to be tested diluted 1:4500 in antibody solution (1% BSA in PBS-T). Bound sea bass immunoglobulins were detected with the mouse anti-sea bass IgM monoclonal antibody WDI3 [68] (1:100 in antibody solution), followed by a horseradish peroxidase-conjugated rabbit anti-mouse antibody (Dako P0260; 1:2000 in antibody solution). Finally, 50 µL of Ultrasensitive TMB solution (ES022, Merck) was added to each well. The reaction was stopped by adding 50 µL of 0.3 M H_2_SO_4_ and the absorbance at 450 nm measured in a BioTek Synergy Mx spectrofluorometer (BioTeK U.S., Winooski, VT, USA). Comparisons between antibody levels in sera from vaccinated and control groups were performed using the Mann–Whitney test and differences were considered significant when *p* < 0.05.

## 5. Conclusions

Evidence presented in this work show that *Phdp* secretes high numbers of OMVs in vitro and in infected hosts. Based on the observation that purified crude OMVs are able to protect *Phdp* cells from the bactericidal activity of fish antimicrobial peptides, we speculate that secretion of OMVs during infection may facilitate the evasion of *Phdp* from the host innate immune system, thereby potentiating the establishment and progression of *Phdp* infections. Vaccination with crude OMVs, rendered non-toxic by removal of the toxin AIP56, induced a specific antibody response and resulted in partial protection against experimental *Phdp* infection. By disclosing new aspects of *Phdp* biology, the findings reported in this work contribute to the understanding of *Phdp* pathogenicity and provide the basis for future works on the pathobiology of this pathogen and on practical applications that may lead to the development of new vaccines able to effectively prevent the negative impacts caused in aquaculture.

## Figures and Tables

**Figure 1 ijms-24-05138-f001:**
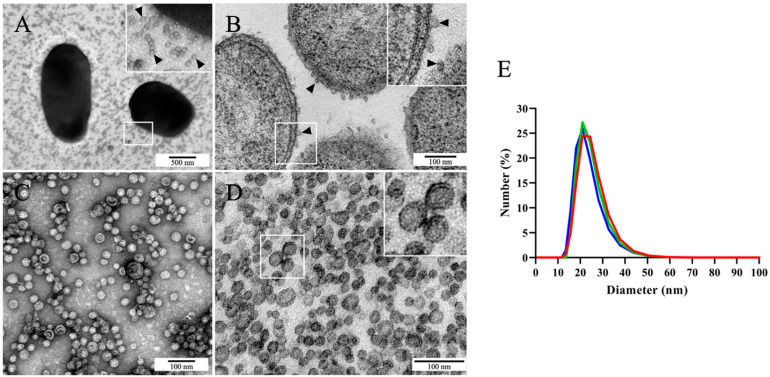
The *Phdp* European strain MT1415 releases large numbers of OMVs in vitro. (**A**) Transmission electron microscopy (TEM) image of negatively stained MT1415 cells surrounded by high number of small-sized spherical vesicles (some vesicles are indicated by arrowheads). Bacteria were cultured on TSA-1, suspended in 1% NaCl, loaded on the grid, and negatively stained with uranyl acetate. (**B**) TEM of an ultrathin section of an MT1415 colony grown on TSA-1, showing several vesicles at the bacterial cells’ surface (some vesicles are indicated by arrowheads). (**C**) TEM of crude OMVs negatively stained with uranyl acetate. (**D**) Ultrathin section of crude OMVs, showing that the vesicles are surrounded by a single bilayer membrane (inset). (**E**) Particle size distribution of crude OMVs analyzed by dynamic light scattering. Each line in the graph corresponds to the results obtained for an independent batch of crude OMVs.

**Figure 2 ijms-24-05138-f002:**
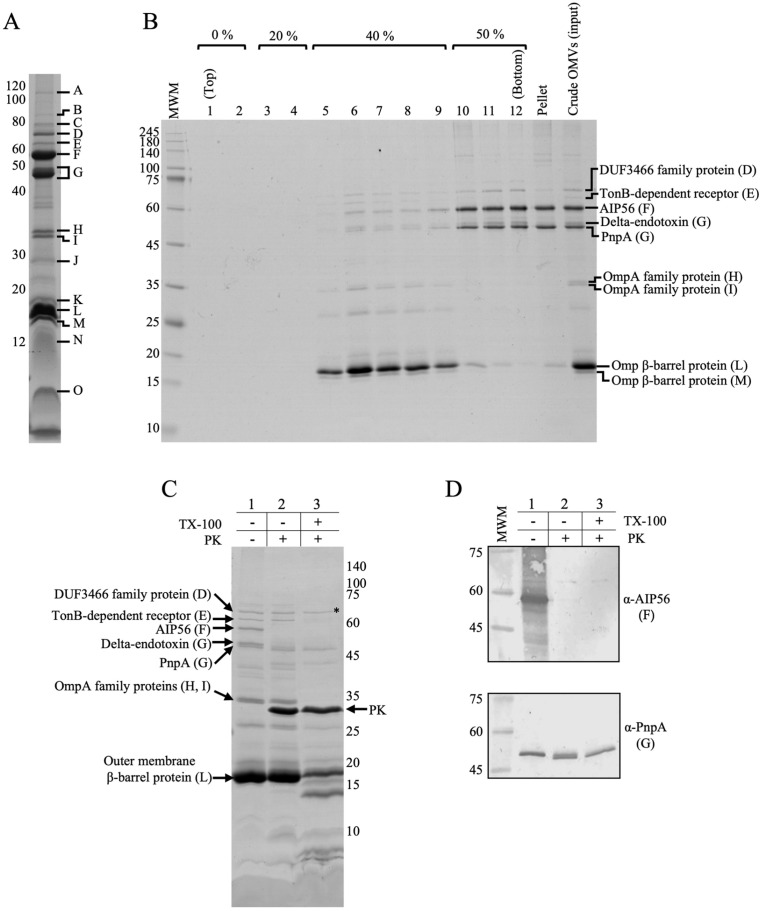
*Phdp* OMVs protein profile. (**A**) Coomassie Blue-stained SDS-PAGE gel of crude OMVs from *Phdp* strain MT1415. The gel was loaded with 8 μL of crude OMVs, which correspond to 30 mL of a late exponential culture (OD_600nm_ of 0.9). The indicated bands correspond to the proteins identified by mass spectrometry (see Table 1). (**B**) The bulk of the T2SS-dependent proteins AIP56 and PnpA present in crude OMVs are not associated with the vesicles. Crude MT1415 OMVs were floated in a discontinuous Histodenz gradient, comprising 50, 40, 20, and 0% Histodenz solution, as indicated. Crude OMVs were loaded at the bottom and fractions were collected from top to bottom. (**C**) AIP56 and PnpA proteins that remain associated with OMVs after floating purification step are located on the outside of the vesicles. Fractions 5–7 of floated OMVs were pooled, either left untreated or treated with proteinase K (PK) in the presence or absence of Triton X-100 (TX-100), and subjected to 15% SDS-PAGE followed by Coomassie staining. The asterisk indicates a protein that is intrinsically resistant to PK digestion. (**D**) Immunoblotting analysis of the samples from panel C using anti-AIP56 and anti-PnpA antibodies. Letters between brackets in panels (**B**–**D**) correspond to the bands indicated in panel (**A**). Numbers at the left or right-side of gels/membranes indicate the position and mass of the molecular weight markers (MWM), in kDa.

**Figure 3 ijms-24-05138-f003:**
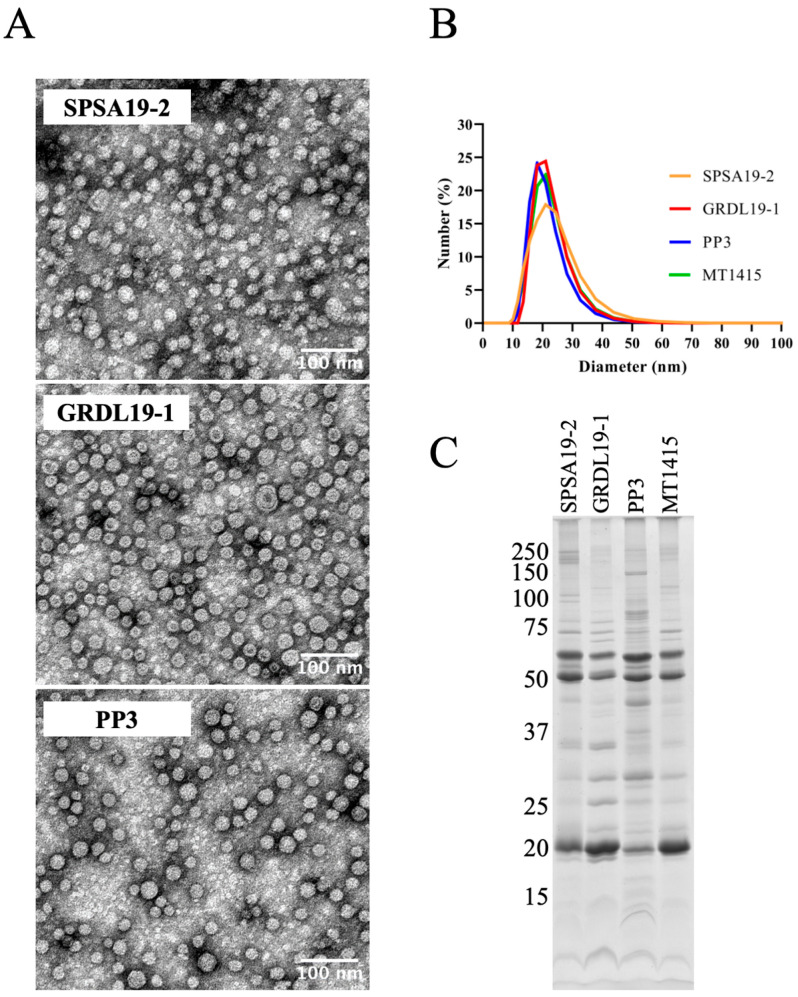
OMVs are produced by the Japanese *Phdp* strain PP3 and by field isolates of *Phdp*. (**A**) TEM of negatively stained crude OMVs from strains SPSA19-2, GRDL19-1, and PP3. (**B**) Dynamic light scattering analysis of the particle size distribution of purified OMVs. Each line corresponds to the average of two independent measurements of the same OMVs preparation. (**C**) SDS-PAGE analysis of crude OMVs. Samples were subjected to SDS-PAGE/Coomassie Blue staining. Each lane contains OMVs equivalent to 30 mL of a late exponential bacterial culture (OD_600nm_ of 0.9). Numbers at the left indicate the position and mass of the molecular weight markers, in kDa.

**Figure 4 ijms-24-05138-f004:**
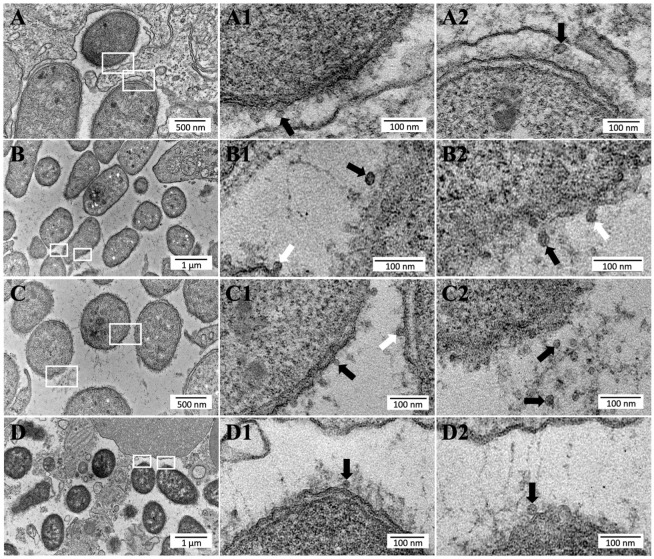
*Phdp* releases OMVs during in vivo infection. Representative TEM images of ultrathin sections of spleens from sea bass experimentally infected with the *Phdp* field isolate GRDL19-1, showing bacterial cells shedding high numbers of OMVs (**A**–**D**). The middle and right pictures (labeled with **1** and **2**) show higher magnifications of the white squared areas. Black arrows indicate fully formed OMVs and white arrows indicate vesicles budding out from the outer membrane of bacterial cells.

**Figure 5 ijms-24-05138-f005:**
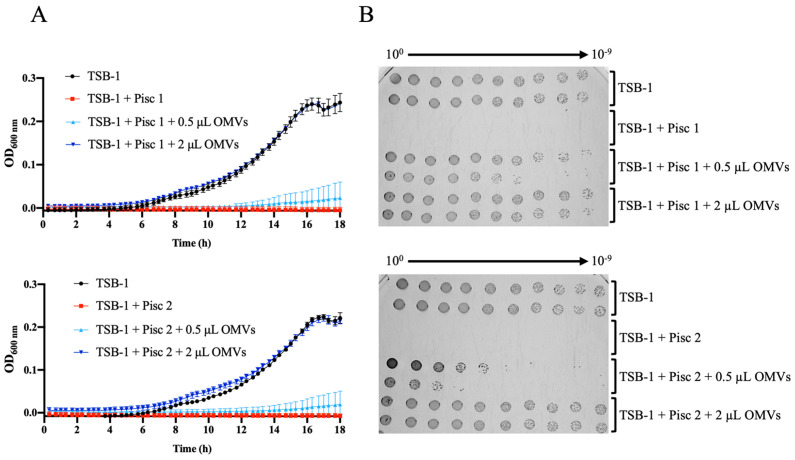
OMVs protect *Phdp* from the microbicidal activity of sea bass antimicrobial peptides. *Phdp* MT1415 was inoculated in TSB-1 (control) or in TSB-1 containing 7.4 or 9.1 µM of sea bass piscidin 1 (Pisc 1) or piscidin 2 (Pisc 2), respectively, supplemented or not with 0.5 or 2 μL of MT1415 crude OMVs, corresponding to a supplementation with 5.9 × 10^13^ or 2.4 × 10^14^ vesicles/mL. Bacterial growth was analyzed by measuring the OD_600nm_ overtime (**A**) and by plating serial 10-fold dilutions of the cultures on TSA-1 plates (**B**). Results shown are representative of three independent experiments.

**Figure 6 ijms-24-05138-f006:**
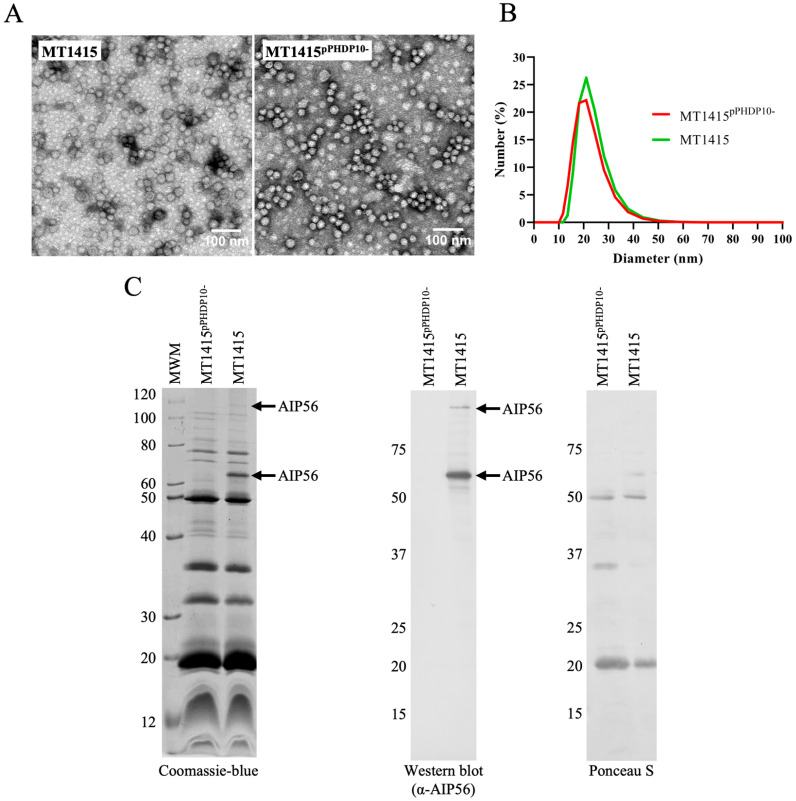
OMVs from MT1415^pPHDP10-^ are similar to MT1415 OMVs but do not contain AIP56. (**A**) TEM of negatively stained crude OMVs from MT1415 and MT1415^pPHDP10-^ strains. Note the similar morphology of the vesicles. (**B**) Crude OMVs from MT1415 and MT1415^pPHDP10-^ have a similar size distribution. The graph shows the size distribution of crude OMVs from the indicated strains determined by dynamic light scattering. (**C**) The protein profile of crude OMVs from the MT1415^pPHDP10-^ strain is similar to the one of crude wild type OMVs, except for the lack of AIP56. Crude OMVs were analyzed by SDS-PAGE/Coomassie staining and Western blotting using an anti-AIP56 antibody. Each lane was loaded with OMVs equivalent to 30 mL (Coomassie Blue) or 7.5 mL (Western blot) of a late exponential culture (OD_600nm_ of 0.9). Numbers at the left-side of gel/membrane indicate the mass of the molecular weight markers, in kDa.

**Figure 7 ijms-24-05138-f007:**
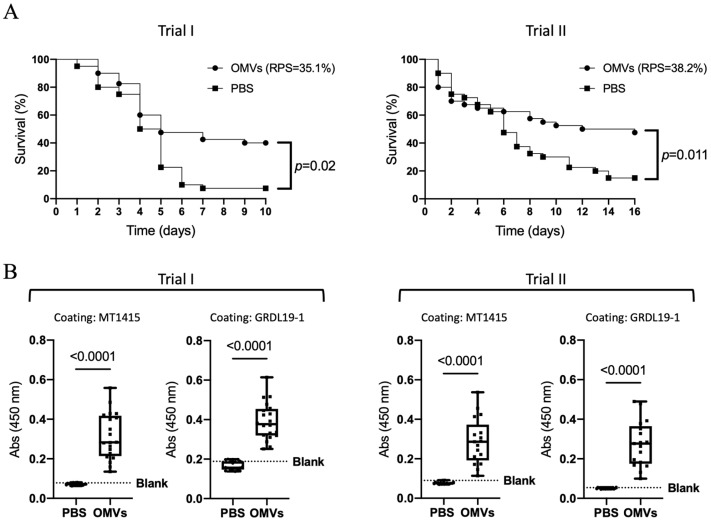
Vaccination with *Phdp* OMVs confers protection against *Phdp* infection. (**A**) Kaplan–Meier survival curves of sea bass vaccinated with MT1415^pPHDP10-^ crude OMVs or injected with PBS (*n* = 40 fish/group) and challenged i.p. with *Phdp* MT1415. Results obtained in two independent trials (I and II) are shown. Group comparisons were performed by the log-rank test and differences were considered significant when *p* < 0.05. (**B**) Fish vaccinated with MT1415^pPHDP10-^ crude OMVs had increased levels of anti-*Phdp* antibodies. Antibodies against UV-killed bacterial cells from *Phdp* strains MT1415 or GRDL19-1 were detected by ELISA. Comparisons between antibody levels in sera from vaccinated fish *(n* = 20 and *n* = 17 for Trials I and II, respectively) and PBS controls (*n* = 20 and *n* = 19 for Trials I and II, respectively) were performed using the Mann–Whitney test and differences were considered significant when *p* < 0.05.

**Table 1 ijms-24-05138-t001:** Identity of the most abundant proteins of crude MT415 OMVs. Bands A–O correspond to the gel bands in Figure 2. Bands D–F were identified by PMF/MALDI-TOF and the remaining bands by LC-MS/Orbitrap.

Band	NCBI Accession Number	Relative Abundance ^1^	MW (kDa) ^2^	Signal Peptide ^3^	Description	Present/Absent in subsp. *damselae* ^4^
A	WP_012954632.1	93.0%	56.2	Sec/SPI	Apoptosis-inducing protein (AIP56)	Absent
B	WP_044175038.1	79.7%	82.7	Sec/SPII	Lipase	Present
C	WP_094461548.1	81.0%	83.1	Sec/SPII	Ig-like domain containing protein	Present
D	WP_044175643.1	na	70.8	Sec/SPI	DUF3466 family protein	Present
E	WP_044179093.1	na	64.4	Sec/SPI	TonB-dependent receptor	Present
F	WP_012954632.1	na	56.2	Sec/SPI	Apoptosis-inducing protein (AIP56)	Absent
G	WP_044178512.1 WP_044176547.1	69.6% 28.8%	55.1 60.2	Sec/SPI Sec/SPI	NlpC/P60 endopeptidase (PnpA) Insecticidal delta-endotoxin Cry8Ea1 family protein	Present Present
H	WP_044174887.1	92.0%	32.7	Sec/SPI	OmpA family protein	Present
I	WP_044174816.1	99.8%	33.7	Sec/SPI	OmpA family protein	Present
J	WP_044176214.1 WP_044174887.1	33.3% 33.2%	42.3 32.7	Sec/SPI Sec/SPI	Omp transport protein OmpA family protein	Present Present
K	WP_044179308.1 WP_094461570.1	61.4% 35.4%	20.5 19.6	Sec/SPI Sec/SPI	TIGR04219 family outer membrane beta-barrel protein Outer membrane beta-barrel protein	Present Present
L	WP_094461570.1	98.0%	19.6	Sec/SPI	Outer membrane beta-barrel protein	Present
M	WP_044178211.1	87.8%	18.8	Sec/SPI	Outer membrane beta-barrel protein	Present
N	WP_081282903.1	64.3%	13.6	Sec/SPII	Glycine zipper 2TM domain-containing protein	Present
O	WP_044177989.1	na	7.0	Sec/SPII	Lpp/OprI family alanine-zipper lipoprotein	Present

^1^ For bands A–C and G–N, in which multiple proteins were identified with statistical confidence, the hits with >20% relative abundance are listed. ^2^ For proteins with a predicted signal peptide, the indicated molecular weight corresponds to the mature form. ^3^ Signal peptides were predicted with SignalP 6.0 (https://services.healthtech.dtu.dk/service.php?SignalP-6.0, accessed on 10 January 2023). ^4^ Presence/absence of the proteins in *Photobacterium damselae* subsp. *damselae* was determined using BLAST—The Basic Local Alignment Search Tool (https://blast.ncbi.nlm.nih.gov/Blast.cgi, accessed on 10 January 2023). na—not applicable.

**Table 2 ijms-24-05138-t002:** *Phdp* strains used in this study.

Strain	Species of Isolation	Geographical Origin	Year
MT1415	*Dicentrarchus labrax*	Italy	Unknown
PP3	*Seriola quinqueradiata*	Japan	Unknown
GRDL19-1	*Dicentrarchus labrax*	Greece	2019
SPSA19-2	*Sparus aurata*	Spain	2019

## Data Availability

All data are provided in the manuscript.

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
