# Peer review of "Characterization and Vaccine Potential of Outer Membrane Vesicles from Photobacterium damselae subsp. piscicida"

_ijms, 2023, doi:10.3390/ijms24065138_

Round 1

Reviewer 1 Report

The article by Teixeira et al. describes the production of OMVs by Photobacterium damselae subsp. piscicida, partially characterizes their content, and proposes its use as the basis for vaccine formulation. It is well-conceived and executed. The results are appropriately discussed and provide very relevant information for combatting the occurrence of photobacteriosis in aquaculture.

I consider that the work is apropiate to be published in its current format.

Author Response

We thank the reviewer for reviewing our manuscript and for the positive comments about our work.

Reviewer 2 Report

The present manuscript characterised the OMVs of Phdp and evaluated the ita potential as vaccine in sea bass. Please state the novelty of the research. There are some issues needed to be explained or fixed as follow.

Title

The naturally released OMV from Phdp cannot be used to prepare vaccine due to its abundance of toxic AIP56. So its inappropriate to describe its vaccine potential, but its mutant.

Introduction

Line36-38: How does Phdp harm these fish species? How many percent of fish are died from it and how important are these fish in both aquaculture industries and wild fish?

Line40-42: Please describe more about the vaccines or other products to prevent or cure the infection currently in use.

Line42: Please list some examples of Phdp outbreaks.

Line68-69: Is there any other vaccine candidate against Phdp infection other than OMV?

Results

Fig. 1: More signals needed in Fig. 1B to see the OMV clearly.

Line240-242: Are the OMVs the same from in vivo and in vitro secretion? The culture convenient are different, and it needs more confirmation. Is it doable to isolate the OMVs directly from fish tissues?

Fig. 5: What’s the concentration of OMV used in the protection assay?

Fig. 6: It would be better to perform a western blotting to confirm the absence of AIP56.

Fig. 7: What’s the difference between Trial I and trial II? What temperature is wild sea bass living in and which season and what temperature does the Phdp infection outbreaks usually occur?

For Fig. 7B, it would be better to collect the blood sample some days after Phdp challenge to study the effect of vaccination upon challenge, not without challenge.

Discussion

A paragraph of conclusion is needed.

Author Response

The present manuscript characterised the OMVs of Phdp and evaluated the ita potential as vaccine in sea bass. Please state the novelty of the research. There are some issues needed to be explained or fixed as follow.

Authors’ reply: We thank the reviewer for reviewing our manuscript and for all comments and suggestions. As you can see below, we addressed all points. We agree that with the changes introduced in response to the reviewer requests (mainly in the Introduction section), the manuscript became more appropriate to the wider readership of IJMS.

Title

The naturally released OMV from Phdp cannot be used to prepare vaccine due to its abundance of toxic AIP56. So its inappropriate to describe its vaccine potential, but its mutant.

Authors’ reply: By “naturally released OMVs”, we intended to refer to OMVs that are spontaneously released by the bacterium (either wild type or mutant strains) without the addition of exogenous inducers (e.g. detergents, sonication) that are often used to induce hypervesiculation in order to increase OMVs yields (see, for example Balhuizen, M.D et al. Outer Membrane Vesicle Induction and Isolation for Vaccine Development. Frontiers in Microbiology 2021, 12). However, to avoid misinterpretations, in the revised version of the manuscript the title was corrected to “Characterization and vaccine potential of outer membrane vesicles from Photobacterium damselae subsp. piscicida”

Introduction

Line36-38: How does Phdp harm these fish species? How many percent of fish are died from it and how important are these fish in both aquaculture industries and wild fish?

Authors’ reply: Information regarding the importance of the mentioned fish species for aquaculture, the percentage of mortality induced by Phdp and the pathology induced in the host were introduced (please see Lines 35-47 of the revised manuscript).

Line40-42: Please describe more about the vaccines or other products to prevent or cure the infection currently in use.

Authors’ reply: Information about the use of antimicrobials in the treatment of Phdp as well as of the concerns associated to their use in aquaculture was introduced (please see Lines 63-68 of the revised manuscript). A sentence clarifying the type of commercially available anti-Phdp vaccines was also included (please see Lines 69-71 of the revised manuscript).

Line42: Please list some examples of Phdp outbreaks.

Authors’ reply: References reporting outbreaks of Phdp in Greece, Australia and Taiwan were added, as requested (please see refs 18-20 in Line 72 of the revised manuscript).

Line68-69: Is there any other vaccine candidate against Phdp infection other than OMV?

Authors’ reply: As stated in the Introduction (Lines 68-69 of the revised manuscript), over the last 30 years, several candidate vaccines against Phdp were proposed, mostly inactivated bacterins sometimes enriched with other components, such as capsular polysaccharide, extracellular products and LPS (reviewed in refs 1 and 17 of the revised manuscript). However, only a few inactivated vaccines have been licensed for aquaculture use and players in the field unofficially report that the conferred protection is not satisfactory. Because of this, several outbreaks of Phdp occur every year in several countries, even in vaccinated stocks, although no official reports are available with this information. However, there are several evidences that confirm the occurrence of Phdp outbreaks. For example, between 2016-2020, our lab had access to ~50 field isolates of Phdp recovered from sea bass and sea bream during outbreaks occurred in Portugal, Spain, Italy and Greece (the list of the isolates can be found in a paper that we published last year: Freitas, I.L. et al. Susceptibility of Sea Bream (Sparus aurata) to AIP56, an AB-Type Toxin Secreted by Photobacterium damselae subsp. piscicida. Toxins 2022, 14, 119) and several studies report the characterization Phdp field isolates (see for example refs 18-20 introduced in Line 72 of the revised manuscript). In this scenario, new and improved vaccines against Phdp are highly needed. We think that our discovery that Phdp produced huge numbers of OMVs in vitro that are able to confer partial protection against experimental infection may be the first step towards the development of an OMVs-based vaccine able to protect fish from Phdp infections.

Results

Fig. 1: More signals needed in Fig. 1B to see the OMV clearly.

Authors’ reply: To facilitate the identification of OMVs, arrowheads were added not only to Fig. 1B, but also to Figure 1A, and the Figure legend was adjusted accordingly.

Line240-242: Are the OMVs the same from in vivo and in vitro secretion? The culture convenient are different, and it needs more confirmation. Is it doable to isolate the OMVs directly from fish tissues?

Authors’ reply: We agree with the reviewer in that this is an important aspect that awaits clarification. However, it is technically challenging to isolate the in vivo produced OMVs pure enough and in sufficient amounts to perform a detailed characterization of the vesicles. Nevertheless, a sentence mentioning the current lack of information regarding this point was added to the Discussion section (Lines 408-410).

Fig. 5: What’s the concentration of OMV used in the protection assay?

Authors’ reply: Information regarding the concentration of the vesicles used in the AMP protection assay was added to the Fig.5 legend.

Fig. 6: It would be better to perform a western blotting to confirm the absence of AIP56.

Authors’ reply: A western blot confirming the absence of AIP56 in the OMVs from the cured strain was added to Fig. 6 and the figure legend was adjusted accordingly.

Fig. 7: What’s the difference between Trial I and trial II?

Authors’ reply: Trial I and trial II correspond to two independent trials (performed separately with two independent lots of fish), as explained in sections 4.13 and 4.14. This is now mentioned in legend for Fig 7 of the revised manuscript. Carrying out two independent trials aimed at ensuring the reproducibility of results.

What temperature is wild sea bass living in and which season and what temperature does the Phdp infection outbreaks usually occur?

Authors’ reply: The European seabass is an eurythermic and euryhaline fish species that tolerates temperatures between 5-28 °C. However, Phdp outbreaks are more prevalent in the summer months and are associated with increased water temperatures (>23ºC). This information was added to the Introduction section.

For Fig. 7B, it would be better to collect the blood sample some days after Phdp challenge to study the effect of vaccination upon challenge, not without challenge.

Authors’ reply: The aim of this analysis was to confirm if the OMVs were able to induce the production of Phdp-specific antibodies. For this, blood had to be collected before challenge, to exclude any possible effect introduced by infection. Nevertheless, we agree with the reviewer that in future studies, it would be interesting to complement this with the analysis of the antibody levels a few days after challenge, to have information about the combined effect of vaccination+infection.

Discussion

A paragraph of conclusion is needed.

Authors’ reply: Following the reviewer’s suggestion, we added a paragraph with the main conclusions of the work. We introduced the conclusions in a separate section (section 5) and not as part of the discussion, to comply with the IJMS manuscript format.